# Ocean fronts and eddies force atmospheric rivers and heavy precipitation in western North America

Xue Liu[1,2,8], Xiaohui Ma [3,4,8✉], Ping Chang [1,2,5], Yinglai Jia[3], Dan Fu [1,2], Guangzhi Xu[6], Lixin Wu [3,4], R. Saravanan[1,5] & Christina M. Patricola-DiRosario [7]

Atmospheric rivers (ARs) are responsible for over 90% of poleward water vapor transport in the mid-latitudes and can produce extreme precipitation when making landfall. However, weather and climate models still have difficulty simulating and predicting landfalling ARs and associated extreme precipitation, highlighting the need to better understand AR dynamics. Here, using high-resolution climate models and observations, we demonstrate that mesoscale sea-surface temperature (SST) anomalies along the Kuroshio Extension can exert a remote influence on landfalling ARs and related heavy precipitation along the west coast of North America. Inclusion of mesoscale SST forcing in the simulations results in approximately a 40% increase in landfalling ARs and up to a 30% increase in heavy precipitation in mountainous regions and this remote impact occurs on two-week time scales. The asymmetrical response of the atmosphere to warm vs. cold mesoscale SSTs over the eddy-rich Kuroshio Extension region is proposed as a forcing mechanism that results in a net increase of moisture flux above the planetary boundary layer, prompting AR genesis via enhancing moisture transport into extratropical cyclones in the presence of mesoscale SST forcing.

[1] International Laboratory for High-Resolution Earth System Prediction, Texas A&M University, College Station, TX, USA. [2] Department of Oceanography, Texas A&M University, College Station, TX, USA. [3] Key Laboratory of Physical Oceanography and Frontiers Science Center for Deep Ocean Multispheres and Earth System, Ocean University of China, Qingdao, China. [4] Qingdao Pilot National Laboratory for Marine Science and Technology, Qingdao, China. [5] Department of Atmospheric Sciences, Texas A&M University, College Station, TX, USA. [6] College of Global Change and Earth System Science, Beijing Normal University, Beijing, China. [7] Department of Geological and Atmospheric Sciences, Iowa State University, Ames, IA, USA. [8] These authors contributed equally: Xue Liu, Xiaohui Ma. ✉email: maxiaohui@ouc.edu.cn

Since the term was coined nearly three decades ago[1], atmospheric rivers (ARs), which are plumes of intense water vapor transport emanating from the atmospheric moisture pool, have been recognized as one of the most important sources of extreme hydroclimate events in the global extratropics, capable of producing torrential rains and floods when making landfall over regions of elevated orography, such as the west coast of North America[1–4]. Some of the most severe river floods in California were associated with ARs[5]. A recent global analysis of ARs' role in driving hydrological extremes found that ARs can contribute not only to extreme floods in many major drainage basins, but also to drought occurrence when ARs are inactive[6]. Therefore, developing and improving the capability of predicting ARs at subseasonal-to-seasonal (S2S) time scales, especially landfalling ARs, can have important implications for water resource management, flood control, and drought relief. Although increasing model resolution in current weather forecast models can lead to forecast bias reduction of overall AR occurrence and intensity, the timing and location of landfalling ARs, as well as their precipitation impact, are notoriously difficult to predict[7–11], underscoring the importance of a better understanding of ARs' dynamics and their predictability sources. There is a concerted, ongoing research effort to understand and quantify predictability and uncertainty in forecasting AR[12–16], including a community-driven project dedicated to evaluate impacts of AR detection algorithms on various science questions[17,18].

Previous studies of ARs' predictability sources have focused on tropical modes of variability at S2S time scales, such as the Madden-Julian Oscillation (MJO)[19,20] and El Niño-Southern Oscillation (ENSO)[21–23]. To the best of our knowledge, none of the published studies explore the potential influence of midlatitude mesoscale SSTs induced by fronts and ocean eddies on ARs, despite the fact that these features along major ocean fronts, such as the Kuroshio Extension (KE) and Gulf Stream (GS), are well known for their impact on the overlying atmosphere, as revealed by high-resolution satellite observations and climate model simulations[24–29]. Evidence is also mounting that the influence of mesoscale SSTs can extend beyond the atmospheric boundary layer, affecting extratropical cyclones (ECs) and midlatitude storm tracks at far distance[30–41]. Given the close association between the occurrence of ARs and ECs[42–47], a natural question to ask is: can mesoscale SSTs influence ARs on S2S time scales, particularly landfalling ARs and associated heavy precipitation events?

In this work, we show that mesoscale SSTs associated with ocean fronts and eddies in the Kuroshio Extension region can exert a remote influence on landfalling ARs and related heavy precipitation along the west coast of North America on S2S time scales. A net increase of moisture flux above the planetary boundary layer (PBL) caused by the presence of Kuroshio mesoscale SSTs prompts AR genesis, which is responsible for the remote influence on the landfalling ARs.

## Results and discussion

**Experiment design.** Two ensembles of twin simulations in the North Pacific sector were conducted using a regional climate model – the Weather Research and Forecasting (WRF) model with a 27-km horizontal resolution (Methods section). Each of the twin-simulations consists of two nearly identical runs with only difference between them being the SST forcing: the control run (hereafter CTRL) was forced with the high-resolution (0.09°) satellite-based MicroWave InfraRed Optimal Interpolated (MW-IR)[48,49] daily SST, whereas the filtered run (hereafter FILT) was forced with the same SST but subject to a low-pass spatial filter to suppress mesoscale features (Methods section). The first

ensemble was based on a set of boreal-winter (6 month) twin simulations from 2002 to 2014 (hereafter seasonal-ensemble or SE), and the second ensemble was based on a set of 2-week twin experiments for a selection of winter cyclone cases (hereafter cyclone-ensemble or CE). The SE experiment was designed to examine the overall impact of mesoscale SST forcing on ARs over the winter season, whereas the CE experiment was designed to further investigate the impact of mesoscale SSTs on ARs during cyclogenesis and development in the KE region. We chose the North Pacific because (1) the largest population exposed to ARs related flood risk is along the west coast of North America[6] and (2) the KE front and eddies generate the strongest mesoscale SST variability in North Pacific (Supplementary Fig. 1) and substantially influence the North Pacific storm track[34,35,50].

**Observed and simulated landfalling ARs and related precipitation.** Figure 1a, b shows the integrated water vapor transport (IVT) averaged over the AR landfalling days in the SE CTRL compared to that derived from the latest European Centre for Medium-Range Weather Forecast (ECMWF) reanalysis (ERA5)[51]. WRF faithfully reproduces the observed landfalling ARs but with a slightly underestimated IVT intensity, due to a higher frequency of landfalling ARs simulated in WRF (Supplementary Fig. 2). Landfalling ARs can produce high precipitation over elevated terrain through orographic lift. This orography-locked precipitation feature appears in the high-resolution dataset based on the Parameter-elevation Regression on Independent Slopes Model (PRISM)[52], and is reproduced remarkably well by WRF (Supplementary Fig. 3). However, because the PRISM dataset covers only the continental U.S., the following observational analysis uses the satellite-based Global Precipitation Measurement (GPM) dataset[53] for heavy precipitation events and the ERA5 reanalysis for landfalling ARs. Although the GPM dataset does not have sufficient resolution to resolve the orography-locked precipitation feature shown in PRISM, it does show a high precipitation concentration along the west coast of North America (Fig. 1f) that corresponds well to the landfalling ARs (Fig. 1b). In fact, the landfalling AR-induced precipitation in GPM shows comparable values to those derived from the SE CTRL (Fig. 1c, f), both of which are considerably greater than the corresponding wintermean (Supplementary Fig. 3), indicating that WRF is skillful in simulating AR-related precipitation. However, the probability density function (PDF) of daily precipitation rate averaged over the west coast of North America (the magenta box in Fig. 1c) shows an overestimation of accumulated precipitation by WRF (Fig. 1d, g). Nevertheless, the fractional contribution (%) of AR-related precipitation to total precipitation exhibits similar distributions between GPM and WRF (Fig. 1e, h). These results give confidence that WRF is capable of realistically simulating AR-induced heavy precipitation along the west coast of North America, thereby providing a basis for further analysis of the influence of mesoscale SST on landfalling ARs and related precipitation statistics.

**Landfalling ARs and heavy precipitation response to mesoscale SSTs.** To understand the relationship between landfalling ARs and heavy precipitation, we selected a subset of landfalling ARs that are concurrent with heavy precipitation events defined as those with daily precipitation rate exceeding the 75th percentile of the area-averaged daily precipitation over the west coast of North America. This subset of landfalling ARs and heavy precipitation events is used in the analyses below. We note that the results do not fundamentally change if extreme precipitation events (exceeding 90th percentile of daily precipitation) are used. Figure 2a, b shows the ensemble-mean landfalling AR IVT

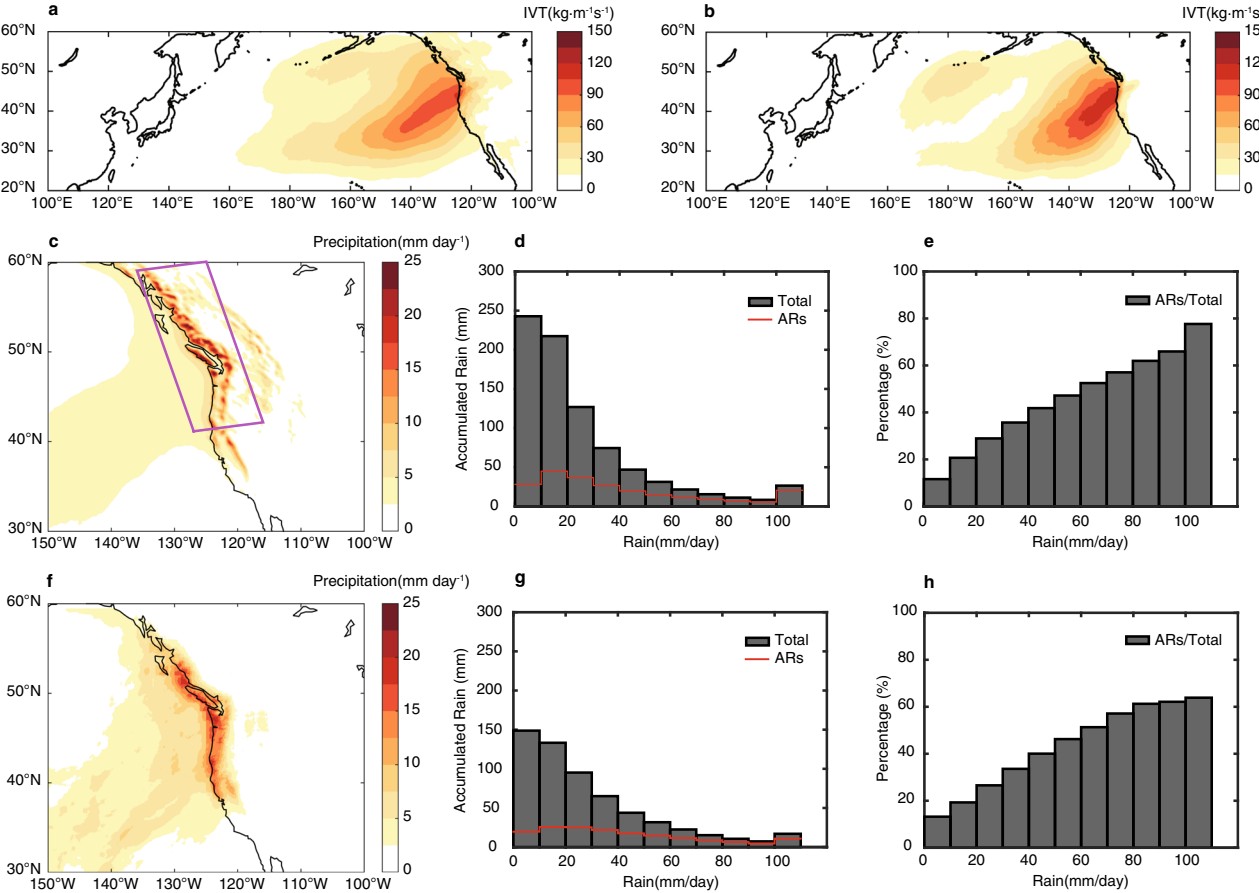

**Fig. 1 Landfalling atmospheric rivers (ARs) and concurrent precipitation along the west coast of North America.** Averaged integrated water vapor transport (IVT) associated with landfalling ARs simulated in seasonal-ensemble control (SE CTRL) (**a**) and in reanalysis data (ERA5, 1979-2017) (**b**) during the winter season (NDJFM). Averaged precipitation associated with landfalling ARs simulated in SE CTRL (**c**) and in satellite-based precipitation data (GPM, 2014-2016) (**f**) during the winter season (NDJFM). Probability density functions (PDFs) of daily precipitation (averaged in the magenta box in Fig. 1c) for total precipitation events (histogram) and those concurrent with landfalling ARs (red line) in SE CTRL (**d**) and in GPM (**g**), respectively. PDFs of fraction contribution (%) of AR-related precipitation to total precipitation in SE CTRL (**e**) and in GPM (**h**), respectively. Averaged IVT (precipitation) is computed as the sum of IVT (precipitation) associated with landfalling ARs divided by the total number of landfalling ARs days.

accumulated over the heavy precipitation days divided by the total number of winter days (150 days) in the SE CTRL and the corresponding value of SE CTRL minus SE FILT, respectively. The reason for using the accumulated rather than averaged IVT is that suppressing of mesoscale SSTs in FILT leads to a significant decrease in both frequency and strength of landfalling ARs. Accumulated IVT takes account of both AR frequency and strength change while averaged IVT undercuts the frequency change. In fact, the total number of landfalling ARs detected in all 65 ensemble members drops from 829 in SE CTRL to 631 in SE FILT. The accumulated IVT of the landfalling ARs is increased by ~40% in SE CTRL compared to SE FILT (Fig. 2b). Figure 2c, d shows the accumulated concurrent heavy precipitation divided by the total winter days and the corresponding difference between SE CTRL and SE FILT. As expected, heavy precipitation over high terrain is most significantly affected by the change in landfalling ARs (Fig. 2d). The presence of mesoscale SST forcing results in up to a 30% increase in heavy precipitation (Fig. 2e) due to the increase of landfalling ARs in the region. Further experiments that separate eddy-induced and front-induced mesoscale SST forcing in the Kuroshio extension region on ARs show eddy-induced mesoscale SST forcing is primarily responsible for landfalling ARs and heavy precipitation response along the west coast of North America (Methods section).

A further support to this finding comes from an analysis on the relationship between strength of mesoscale SST forcing and strength of landfalling ARs and heavy precipitation response. Because of the shortness of the record, we simply grouped the 13 years (2002–2014) of SE CTRL into two 4-year sets based on the strength of mesoscale SST forcing and compared the landfalling ARs and heavy precipitation between these two sets (Methods section). The results show that the set with stronger mesoscale SST forcing produces stronger landfalling AR and precipitation response (Fig. 3). However, a similar analysis applied to the PRISM did not yield a statistically significant precipitation response. This may not be surprising because (1) PRISM only covers a limited area that is south of the most significant precipitation response to mesoscale SSTs and (2) the data records are relatively short and sample size is relatively small.

This modeling result finds support from ERA-Interim reanalysis in which the SST forcing was switched from a 1° low-resolution to a 0.5°-and-finer high-resolution observation dataset before and after 2002[54] (Methods section and Supplementary Fig. 4a). Contrasting the landfalling AR IVT along the west coast of North America between the periods of 2002–2017 and 1986–2002 indicates an increase of the IVT in the former (Supplementary Fig. 4c, e). A similar increase is noted when comparing the landfalling AR IVT between ERA-Interim and the

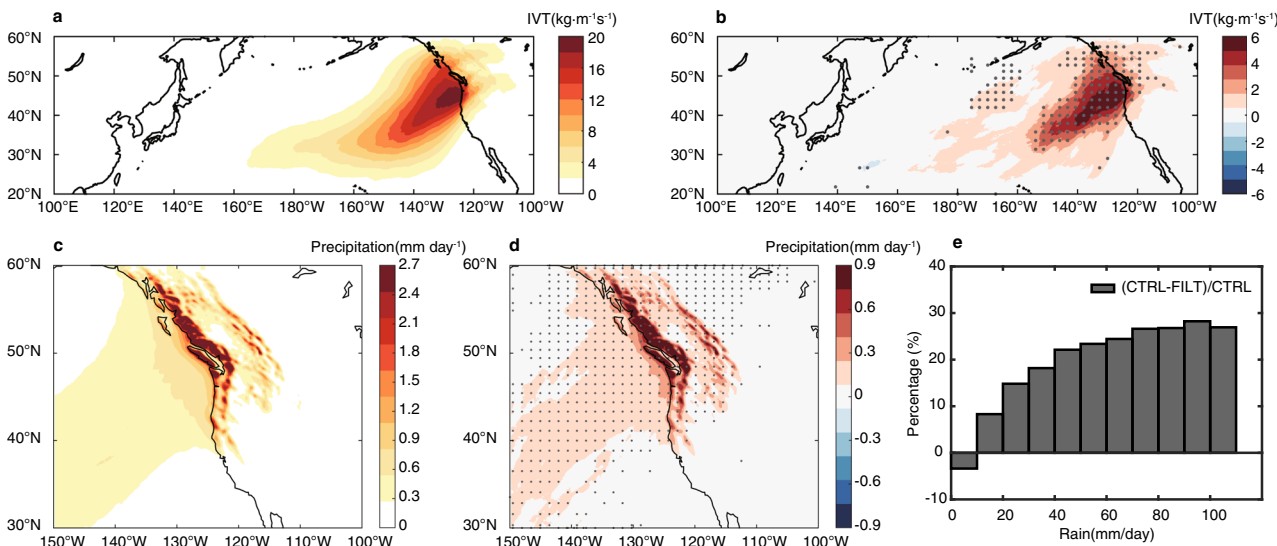

**Fig. 2 Response of landfalling atmospheric rivers (ARs) and heavy precipitation to mesoscale sea surface temperatures in the seasonal ensemble experiment.** Landfalling AR integrated water vapor transport (IVT) summed over the heavy precipitation days and then divided by the total number of days (150 × 65) in seasonal-ensemble control (SE CTRL) (**a**) and the corresponding difference between SE CTRL and filtered (FILT) (**b**). **c**, **d** same as **a** and **b**, but for precipitation. Probability density function (PDF) of relative difference of precipitation concurrent with landfalling ARs between SE CTRL and FILT in reference to SE CTRL (the red curve in Fig. 1) (**e**). Heavy precipitation events are defined as area-averaged (magenta box in Fig. 1c) daily precipitation events exceeding the 75th percentile of the value. The difference above 95% confidence level based on a two-sided Student's *t* test is shaded by gray dots.

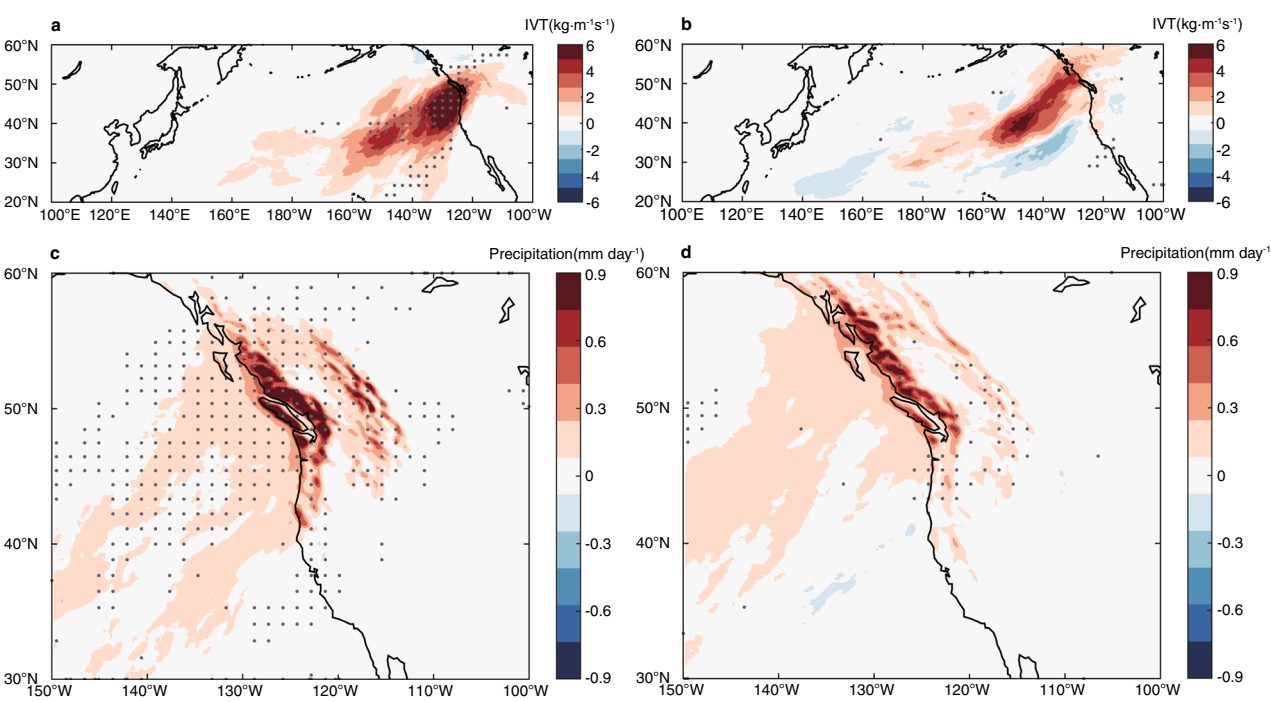

**Fig. 3 Relationship between atmospheric rivers (ARs)/precipitation response strength and sea surface temperature forcing strength in the seasonal ensemble. a**, **c** same as Fig. 2b, d, but for four strong mesoscale SST forcing cases. **b**, **d**, same as Fig. 2b, d, but for four weak mesoscale SST forcing cases.

latest ERA5 reanalysis product where an even higher resolution SST forcing (0.25°) was employed[55] from 1979 to 2006 along with a higher model resolution (Methods section and Supplementary Fig. 4b, d, f). These observation-based analyses, albeit not a proof, provide supporting evidence for the model findings.

The CE experiment allows us to further address the question of whether mesoscale SST forcing along the KE front is responsible for the change in landfalling ARs and heavy precipitation. The CE ensemble consists of 568 winter cyclone cases selected from the

SE CTRL such that they all passed through the KE region (Methods section). Therefore, the majority of ARs generated in this experiment are closely related to these cyclones that pass over the mesoscale SST forcing along the KE. Remarkably, despite the short integration period, the landfalling AR IVT is significantly increased in CE CTRL compared to CE FILT (Supplementary Fig. 5), leading to a significant increase in the precipitation amount due to the presence of mesoscale SST features (Fig. 4a, b). The 2-week mean fractional increase of precipitation (>40 mm

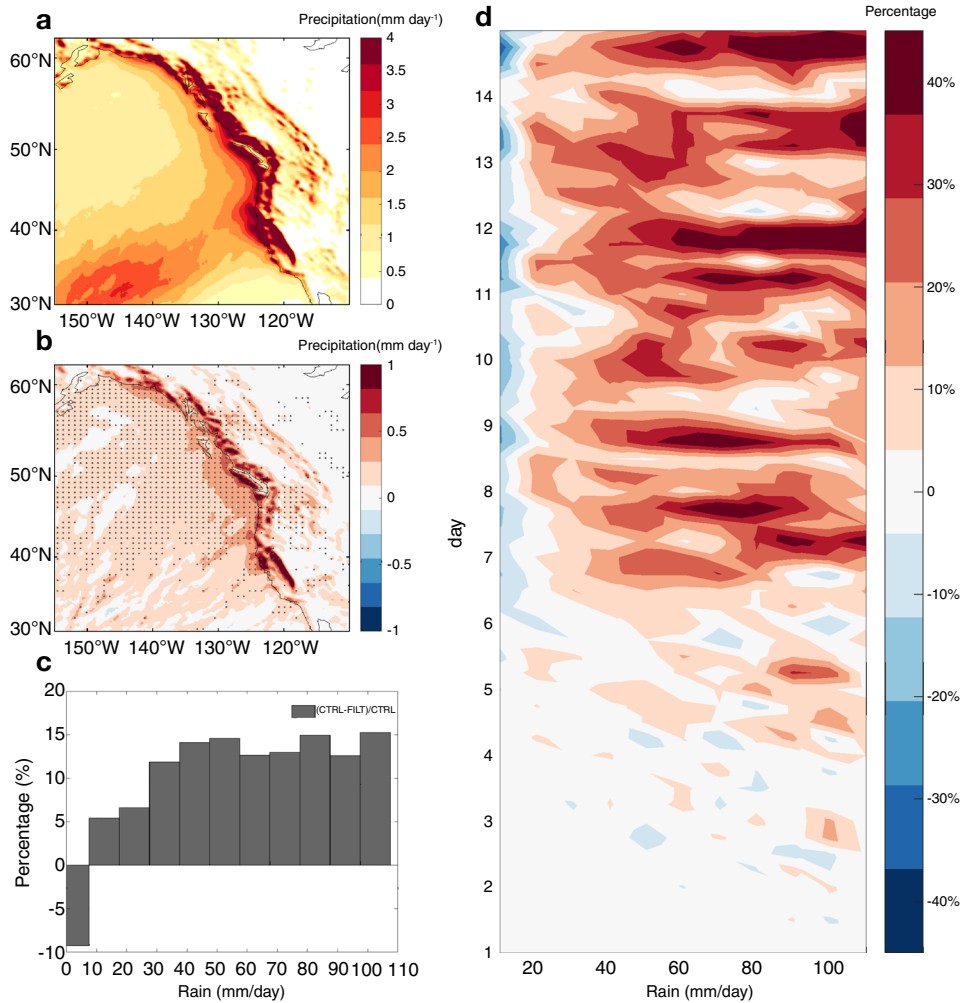

**Fig. 4 Response of heavy precipitation along west coast of North America to mesoscale sea surface temperature forcing in the cyclone ensemble.**
Two-week mean heavy precipitation associated with landfalling atmospheric rivers (ARs) in cyclone-ensemble control (CE CTRL) (**a**) and the corresponding difference between CE CTRL and filtered (FILT) (**b**). Two-week mean (**c**) and time evolving (**d**) probability density function (PDF) of relative difference of 6-hourly precipitation (averaged in the magenta box in Fig. 1c) concurrent with landfalling ARs between CE CTRL and FILT in reference to CE CTRL. The mean precipitation is computed as the sum of landfalling AR-induced precipitation over the heavy precipitation days divided by the 2-week simulation period. The difference above 95% confidence level based on a two-sided Student's $t$ test is shaded by gray dots.

day$^{-1}$) in CE CTRL compared to CE FILT can reach 15% (Fig. 4c). More interestingly, the time-evolving pdf of the fractional precipitation change along the west coast of North America reveals a delayed response: negligible change within the first 4 days, but a significant increase of up to 40% for the precipitation higher than 40 mm per day afterwards (Fig. 4d). This indicates that the influence of mesoscale SST forcing can occur on weekly time scales, highlighting the need for future investigations on whether the mesoscale SSTs can potentially affect the predictability of AR-related heavy precipitation events along the west coast of North America on S2S time scales.

**Mechanism of mesoscale SST forcing ARs.** The delayed precipitation response to the mesoscale SST forcing points to a remote forcing mechanism on landfalling ARs that originates in the eddy-rich KE region. We hypothesize that the key process involved in this remote mechanism lies in the strong influence of mesoscale SST features along the KE (Supplementary Fig. 1) on the net moisture supply to the developing cyclones over this region. This effect is particularly strong following each initial cyclone (selected to initialize the twin ensemble runs) passing

through the KE region. In the wake of the initial cyclone's cold front, cold and dry air descends over the KE region. Over warm mesoscale SSTs the atmosphere is destabilized, intensifying vertical mixing, and resulting a strong upward vertical moisture flux that pumps moisture out of the PBL. Over cold mesoscale SSTs, however, such a moisture pumping does not occur, because the atmosphere is more stable there. As a result, there is a net increase of moisture above the PBL (Fig. 5a, b). A recent modeling study shows a similar moisture increase when ocean eddy-induced SSTs are included in a set of WRF simulations[33]. This increase of moisture supply from the PBL over the KE acts to moisten the precyclone environment for the next developing cyclone. As such, when the next cyclone develops over the KE, the airflow within the warm sector of the cyclone, known as the feeder airstream[44], can transport more moisture into the cyclone. A branch of this feeder airstream feeds to the warm conveyor belt ascent, contributing to cyclone precipitation, while another branch turns away from the cyclone, exporting moisture from the cyclone to form AR[44]. Thus, the ability of mesoscale SSTs to moisten precyclone environment over the KE region can lead to increase in AR IVT. This mechanism also explains the delayed precipitation response along the west coast of North America, because the

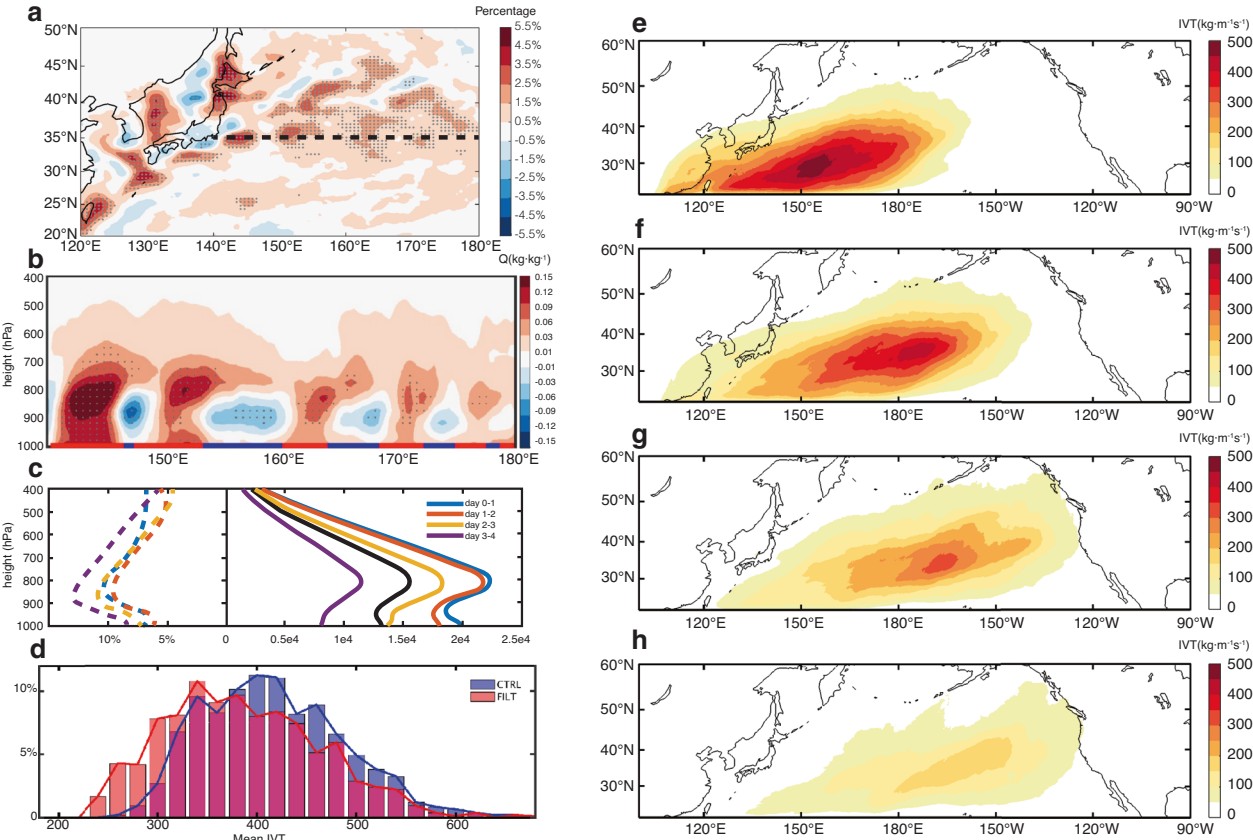

**Fig. 5 Mechanism of Kuroshio mesoscale sea surface temperature (SST) forcing atmospheric rivers (ARs).** Percentage difference of 2-week mean water vapor mixing ratio (Q) at 800 hPa between cyclone-ensemble control (CE CTRL) and filtered (FILT) relative to CE CTRL (**a**). Difference of 2-week mean water vapor mixing ratio (Q) along the Kuroshio Extension (dashed line in **a**) between CE CTRL and FILT (**b**). The heavy line at the bottom of **b** indicates warm (red) and cold (blue) mesoscale SST anomalies. Vertical profile of daily-mean Q anomalies carried by all the ARs within each daily interval, 0–1 day (blue), 1–2 day (red), 2–3 day (yellow), and 3–4 day (purple) following ARs' evolution in CE CTRL (solid lines) and the corresponding fractional difference between CE CTRL and FILT relative to CE CTRL (dashed lines) (**c**). The thick black line shows the vertical profile of daily mean Q anomalies carried by all ARs detected in reanalysis data (ERA5) averaged over 10 boreal winter seasons (NDJFM) from 2008 to 2017. Probability density functions (PDFs) of ARs when they are first formed in CE CTRL (blue) and FILT (red) (**d**). Daily mean integrated water vapor transport (IVT) of all ARs detected within each daily interval, 0–1 day (**e**), 1–2 day (**f**), 2–3 day (**g**), and 3–4 day (**h**) following ARs' evolution in CE CTRL. The daily mean values in **c** and **e–h** are all derived from 6-hourly model output. The daily mean Q anomalies in **c** are computed as the sum of area accumulated Q anomalies within ARs detected on corresponding days divided by the total number of corresponding days. The daily mean IVT in **e–h** is computed as the sum of IVT of all ARs detected on corresponding days divided by the total number of corresponding days. The difference above 95% confidence level based on a two-sided Student's t test is shaded by gray dots.

initial cyclone does not produce major differences in ARs and associated heavy precipitation due to the small difference in the precyclone environment between CTRL and FILT during the initial stage.

The impact of mesoscale SSTs on cyclone-related AR IVT generation is revealed by a composite analysis (Fig. 6). Despite of the small differences in intensity and structure of the composite cyclone between CE CTRL and CE FILT (Fig. 6a, b), there is a significant increase of AR IVT and precipitation in the warm sector of the composite cyclone when mesoscale SSTs are present in CE CTRL (Fig. 6c–f). Since the majority (>85%) of ARs is associated with ECs in the simulations, we can conclude that the cyclone-related AR IVT difference is largely responsible for the total AR difference, including the landfalling AR difference, between CTRL and FILT. It indicates that the presence of mesoscale SSTs can enhance AR genesis by increasing IVT associated with ECs even though cyclone intensity remains unchanged.

The enhanced moisture supply above PBL over the KE region is also observed in ERA-Interim by contrasting the periods between high- and low-resolution SST forcing (Supplementary Fig. 4g, h). As shown by both WRF and the reanalysis, maximum moisture anomalies carried by the ARs occur at ~800 hPa, and in WRF the value is ~10% higher in CE CTRL than in CE FILT during the first day of ARs (Fig. 5c). This enhanced moisture supply promotes generation of stronger ARs in the region, as shown by the AR genesis PDF (Fig. 5d) that displays a marked shift towards higher IVT values in CE CTRL compared to CE FILT. Since the stronger ARs are more likely to survive the journey across the Pacific, the fractional difference of AR moisture content between CE CTRL and FILT is expected to increase (Fig. 5c) as ARs move westward, eventually making landfall along the west coast of North America and impacting rainfall in the region (Fig. 5e–h). The time scale associated with this remote mechanism is ~4–5 days, consistent with the result in Fig. 4d. This mechanism of enhanced vertical moisture transport by warm mesoscale SSTs is further tested for its robustness by using different PBL schemes in WRF. The results show that all the schemes produce a net increase of water vapor content above the PBL in response to mesoscale SST forcing (Methods section and

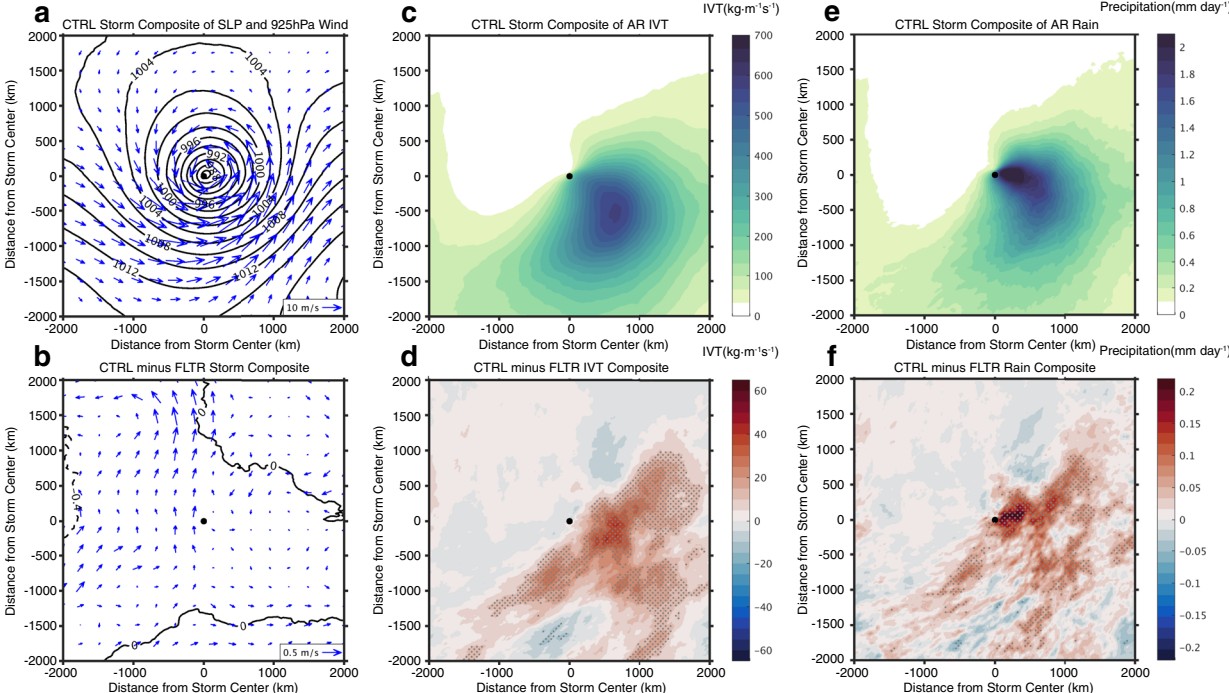

**Fig. 6 Composite of extratropical cyclones in the cyclone ensemble experiment.** Composite of sea level pressure (SLP, contours, hPa) and 925hPa wind (vector, ms$^{-1}$) of all identified extratropical cyclones in cyclone-ensemble control (CE CTRL) (**a**), and the corresponding differences between CE CTRL and filtered (FILT) (**b**). The black dot indicates the center of composite cyclone. **c**, **d** same as **a** and **b**, but for composite of AR integrated water vapor transport (IVT) associated with extratropical cyclones. **e**, **f** same as **a** and **b**, but for composite of AR-induced precipitation. The difference above 95% confidence level based on a two-sided Student's *t* test is shaded by gray dots.

Supplementary Fig. 6). The overall percentage of water vapor increase in WRF experiments is in line with that in ERA-Interim (Supplementary Fig. 4h), indicating the modeling results are not sensitive to WRF PBL physics parameterizations.

**Supporting evidence from global simulations.** The above results are also supported by a similar twin experiment using a high-resolution global model – a version of the Community Atmosphere Model (CAM) at 25 km horizontal resolution with (CTRL) and without (FILT) mesoscale SST forcing in both the KE and GS Extension region (Methods section). The CAM results not only show a consistent AR and precipitation response with WRF to KE mesoscale SST forcing in the North Pacific, but also reveal an increase in landfalling ARs and precipitation along the European west coast to GS mesoscale SST anomalies (Supplementary Fig. 7). Given differences in model physics and numerics between WRF and CAM, the consistency between these modeling results further points to the robustness of the findings.

Collectively, these numerical simulations and observations support the hypothesis that mesoscale SST forcing associated with oceanic fronts and eddies in western boundary current regions substantially influences landfalling ARs and associated heavy precipitation on S2S time scales. It indicates that the common practice of using non-eddy-resolving monthly SST as forcing in atmosphere-only models can lead to underestimates of AR-induced heavy precipitation, even for high-resolution atmospheric models. Although further prediction experiments are required to quantify the impact of mesoscale SSTs on predicting ARs, this study does point to the potential importance of including SST mesoscale forcing in prediction models.

## Methods
**Seasonal ensemble.** The Weather Research and Forecasting (WRF) model[56] was configured over the North Pacific ([3.6°N 66°N], [99°E 270°E]) with 27 km

horizontal resolution and 30 vertical levels. The initial and lateral boundary conditions were derived from 6-hourly NCEP-DOE AMIP-II reanalysis (NCEP-II)[57]. The experiment contains 13 boreal winter twin simulations from 2002 to 2014, each of which has an ensemble of 5 CTRL and FILT runs, respectively, initialized on October 1 with slightly different states and integrated for 6 months. The total ensemble size is 65 winter seasons for each of the twin simulations. The low-pass spatial filter applied to the daily SST forcing in the FILT ensemble is a 2-D Loess filter with a 15° × 5° cut-off wavelength[58]. Only the last 5 months were analyzed, with the first month disregarded as model spin-up. Detailed parameterization schemes were used in the model configuration including Lin et al.'s scheme for microphysics, RRTMG and Goddard scheme for longwave and shortwave radiation, a Noah land surface scheme, YSU scheme for planetary boundary layer, and the Kain-Fritsch scheme for convection following a previous study[30].

**Cyclone ensemble.** The same WRF model configuration was used for this set of twin simulations. A total of 568 winter cyclone cases that pass through the KE region were selected from the SE CTRL. A pair of 14-day simulations – one CTRL and one FILT – were made for each cyclone case, initialized with the restart file and lateral boundary conditions from SE CTRL, but different SST forcing. CE CTRL uses the original SST from SE CTRL, but the SST in CE FILT is subject to the same Loess filter used for SE FILT.

**PBL sensitivity experiments.** The same WRF model with different PBL and surface-layer schemes was configured to test the sensitivity of water vapor response to different physics parameterizations. The sensitive experiments contain four twin simulations in 2007/8 winter season, each of which has an ensemble of 10 CTRL and FILT runs, respectively. Based on previous studies[30], 10 ensemble members are sufficient to identify a significant atmospheric response to mesoscale SST forcing in WRF because the use of identical lateral boundary conditions acts to reduce the atmospheric internal variability. All the settings in these experiments were the same as WRF SE except that different PBL and surface layer schemes were used, including YSU, MYNN, UW PBL with MM5 surface layer scheme, and MYJ PBL with Eta surface layer scheme[56].

**Global ensemble.** The global climate model used is a version of the Community Atmospheric Model Version 5 (CAM5) at 0.23° horizontal resolution with prescribed SST and sea ice derived from daily 0.25° NOAA Optimum Interpolation SST and ICE (OISSTV2). Similar to the WRF design, the global CAM simulations contain 13 boreal winter twin simulations from 2002 to 2014, each of which only has an ensemble of 2 CTRL and FILT runs, respectively, initialized on December 1

with slightly different states and integrated for 3 months. In CTRL, the 0.25° OISSTV2 was retained. In FILT, a 5° × 5° (4° × 4°) boxcar low-pass filter was applied to OISSTV2 in the Kuroshio (Gulf Stream) Extension region, respectively. The different filtering window width is based on the typical length scale of mesoscale SSTs in these two regions[59–61].

**ARs detection and tracking.** To identify and track ARs, we adopted a widely used AR detection approach[3] that searches for long narrow region of IVT anomalies of values exceeding a threshold and IVT anomalies are referred to temporal mean[62,63]. Specifically, ARs are identified as continuous areas of IVT $\left(\frac{1}{g}\int_{1000hPa}^{300hPa}|\vec{V}Q|dp\right)$ anomalies that exceed $250\,\mathrm{kg\,m^{-1}s^{-1}}$, where $\vec{V}$ is winds and $Q$ is water vapor mixing ratio. For SE, the anomalies were derived from daily mean IVT subtracting climatological winter season (NDJFM) mean of all 65 ensemble members. For CE, the anomalies were derived from 6-hourly IVT subtracting 2-week mean of all 568 ensemble members. For the global CAM simulations, the anomalies were derived from daily mean IVT subtracting winter season (DJF) mean of all ensemble members. The outer edge of an AR is defined by a closed IVT contour of $250\,\mathrm{kg\,m^{-1}s^{-1}}$. The length of an AR must be longer than 2000 km while its width must be narrower than 1000 km. ARs were designated as landfalling if the outmost contour insects coastline. The center of an AR is defined as the geometric center of the IVT contour. Coherent AR object is stitched from a Lagrangian tracking approach[64] to form an AR trajectory. We tested a different AR detection algorithm using integrated water vapor and the tracking results by including or not including a temporal requirement that tracked ARs must last for at least 18 h, the results showed no significant effects on the conclusion of this study.

**Definition of heavy precipitation events.** Heavy precipitation events along the west coast of North America are defined as area-averaged (magenta box in Fig. 1c) daily precipitation events exceeding the 75th percentile daily precipitation rate. To test the robustness of the results, the analysis was repeated using extreme precipitation events (exceeding 90th percentile daily precipitation rate) and results consistently show higher precipitation associated with landfalling ARs in SE CTRL than FILT. Heavy precipitation events were used because it allows for a larger sample size, increasing the robustness of the results.

**Relationship between ARs/precipitation response strength and mesoscale SST forcing strength.** ARs/precipitation response – mesoscale SST forcing relationship is examined by first dividing the 13 years (2002–2014) of SE simulations into two sets based on the strength of mesoscale SST forcing and then contrasting ARs/precipitation responses between the two sets. The strength of mesoscale SST forcing is measured by area-averaged mesoscale SST anomaly variance in the Kuroshio Extension region ([20°N 45°N, 120°E-180°E]). The set with strong (weak) mesoscale SST forcing includes top 4 highest (lowest) mesoscale SST variance winters with a total of 20 ensemble members. The differences of landfalling ARs IVT and precipitation between the two sets are shown in Fig. 3.

**Comparison of ARs and water vapor in ERA-interim and ERA5.** The SST forcing in ERA-Interim was switched from low resolution (1°) to high resolution (0.5° and finer) before and after 2002, while SST in ERA5 has an even higher resolution of 0.25° from 1979 to 2006[54,55]. Comparison of ERA-Interim before and after 2002 is similar to the comparison between SE CTRL and FILT, but based on two 16-year periods of 1986–2001 and 2002–2017. For ERA-I and ERA5 comparison, the period of 1979–2006 was used.

**Extratropical cyclones detection and EC/AR composite.** A widely used winter storm detection approach[65] was used to identify and track ECs. Centers of ECs were first identified by sea level pressure (SLP) minima within a closed contour, with an additional requirement of at least 1 hPa increase of SLP within 5° of the center. In total, 400 hPa temperature was used to detect and eliminate those with a warm core so that the identified ECs are distinct from tropical cyclones. Candidates are then stitched in time to form paths, with a maximum distance of 6° between them. The identified ECs must have a duration of at least 2 days and a traveling distance of 10°. Following a recent study[66], if both an AR and EC co-exist within a 4000-km × 4000 km box centered at EC center, the related field including SLP, IVT, and precipitation are used in the composite analysis. The detection of ECs and composite analysis were performed in the whole North Pacific region([150°E–240° E, 20°N–60°N]).

**Relative importance of ocean eddies vs. SST fronts in driving the AR response.** Mesoscale SST anomalies that affect ARs can be induced by both ocean eddies and fronts. To separate the effect of ocean eddies vs. SST fronts in driving ARs, additional ensembles of WRF simulations were performed for 2007–2008 winter season with each ensemble containing 10 simulations. As with the other simulations, only SST forcing is different for different ensembles. The first set of ensembles is a repeat of SE-CTRL and SE-FILT (hereafter referred to as the front-eddy (F-E) ensemble where both eddy- and front-induced mesoscale SST forcing is included as shown in Supplementary Fig. 8a). The second set of ensembles is identical to the first

set except that CTRL and FILT were forced by climatological monthly mean MW-IR SST and the corresponding filtered SST climatology, respectively (hereafter referred to as the front-only (F-O) ensemble where front-induced mesoscale SST forcing is included as shown in Supplementary Fig. 8b). The third set of ensembles is also similar to the first set except that in FILT only eddy-induced SST anomalies were removed but the SST front was retained so that the difference between CTRL and FILT reflects only the influence of eddy-induced SST on ARs (hereafter referred to as the eddy-only (E-O) experiment where only eddy-induced mesoscale SST forcing is included as shown in Supplementary Fig. 8c).

In F-E, the landfalling AR and precipitation differences between CTRL and FILT show similar patterns to those of the 13 winter season simulations (Fig. 2b) except for a slight southward shift. Overall, the presence of mesoscale SSTs leads to an increase of landfalling AR IVT and heavy precipitation along the west coast of North America (Supplementary Fig. 8d, g). In contrast, in F-O where only SST front is present, the landfalling ARs and associated heavy precipitation along the west coast of North America show a decrease (Supplementary Fig. 8e, h), which is opposite to the result in F-E. This suggests that the increase in landfalling ARs and precipitation can only be attributed to eddy-induced mesoscale SST forcing, which is confirmed by E-O (Supplementary Fig. 8f, i). Not only the AR response in E-O has the same sign as in F-E, but also it is stronger. These results suggest it is the ocean eddy-induced mesoscale SST forcing that is primarily responsible for the increase in landfalling ARs and associated heavy rainfall along the west coast of North America. It is interesting to note that a stronger Kuroshio SST front can lead to a decrease (not increase) in the landfalling AR and heavy rainfall which is intriguing and deserves future investigations.

**Significance test.** A Student's $t$ test is applied for a given variable when comparing the difference between CTRL and FILT, assuming each of the detected ARs is an independent sample. Data value in regions without ARs occurrence is set to zero.

## Data availability

SST and lateral boundary conditions were prescribed to WRF from the daily satellite microwave and infrared (MWIR) SST (http://www.remss.com/measurements/sea-surface-temperature) and 6-hourly NCEP-DOE AMIP-II reanalysis (https://www.esrl.noaa.gov), respectively. Observed ARs were derived from the European Centre for Medium-Range Weather Forecasts (ECMWF) ERA-Interim and ERA5 reanalysis (https://www.ecmwf.int/en/forecasts/datasets). Rainfall derived from Parameter-elevation Regression on Independent Slopes Model (PRISM) and Global Precipitation Measurement (GPM) satellite observations were used to compare with WRF simulations. The model output data in this study are available from the corresponding author upon request.

## Code availability

The Weather Research and Forecasting (WRF) model and the Global Community Atmospheric Model (CAM) are developed by the National Center for Atmospheric Research (NCAR). Model code is available at (https://www.mmm.ucar.edu/weather-research-and-forecasting-model and http://www.cesm.ucar.edu). The analysis code used in this study is available from the corresponding author upon request.

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

## Acknowledgements

This research is supported by the Natural Science Foundation of China 41776013 and U.S. National Science Foundation Grant AGS-1462127, National Key R&D Program of China (2017YFC1404100, 2017YFC1404101) and Natural Science Foundation of China 41776009, the U.S. Department of Energy under Award Number DE-SC0020072 and the U.S. Department of Commence under Award Number NA20OAR4310409. Y.J. acknowledges the support from the Natural Science Foundation of China (41975065). X.L. acknowledges the support from the China Scholarship Council. C.M.P. acknowledges support from the U.S. Department of Energy, Office of Science, Office of Biological and Environmental Research, Climate and Environmental Sciences Division, Regional & Global Climate Modeling Program, under Award Number DE-AC02-05CH11231. We thank the Texas A&M Supercomputing Facility and the Texas Advanced Computing Center (TACC), the Center for High Performance Computing and System Simulation at Qingdao Pilot National Laboratory for Marine Science and Technology (QNLM) for providing high performance computing resources that contributed to the research results reported in this paper. This is a collaborative project between the Ocean University of China (OUC), Texas A&M University (TAMU) and the National Center for Atmospheric Research (NCAR) and completed through the International Laboratory for High Resolution Earth System Prediction (iHESP) – a collaboration among QNLM, TAMU and NCAR.

## Author contributions

X.L. and X.M. made equal contribution to the model simulations and data analyses. X.L. conducted and analyzed the prediction ensemble simulations, while X.M. conducted and analyzed the seasonal ensemble simulations. Additionally, X.M. carried out the observed AR and precipitation analyses. X.L. conducted the extropical cyclones related analyses. P.C. conceived the original idea and provided interpretations of the results. P.C., X.M., and X.L. co-directed the research and co-wrote the manuscript. Y.J. conducted and analyzed the global CESM simulations. D.F. and G.X. assisted with AR tracking. All authors contributed to improving the manuscript.

## Competing interests

The authors declare no competing interests.
