## [Peer Review File · Nature Communications]

REVIEWERS' COMMENTS

Reviewer #1 (Remarks to the Author):

Re-review

Title: Ocean Fronts and Eddies Remotely Forcing Atmospheric Rivers and Heavy Precipitation

Authors: Xue Liu, Xiaohui Ma, Ping Chang, Yinglai Jia, Dan Fu, Guangzhi Xu, Lixin Wu, R. Saravanan, Christina M. Patricola

The authors have satisfied my review points and I can recommend publication. However, I encourage them to also consider the following two additional comments before publication.

1. Response to request for PRISM data inclusion, Lines 122-123 original manuscript: Thank you for this explanation. Something similar should be noted in the paper itself because I think this will be a common question amongst most readers.

2. Additional experiments as a response to Reviewer 2: I find the difference between the eddy-only and front-only experiments compelling and very intriguing! This is indeed a valuable addition to this work and I strongly recommend that it is a) included in the extended figures, in particular figure R7 and b) added as a subsection in the Methods section as supplemental material focused on isolating fronts vs eddies. I look forward to analysis on why the front-only experiment lead to a decrease in IVT and precipitation!

Reviewer #3 (Remarks to the Author):

This study examines the role of mesoscale SST eddies in the North Pacific on AR frequency and intensity - including landfalling IVT and precipitation. It tests the hypothesis that these mesoscale SST features impart a non-linear impact on the transport of moisture out of the PBL, that leads to greater moisture export out of the boundary layer into the free atmosphere in the region of cyclone development leading to increased AR frequency and intensity. They test this hypothesis with a number of modeling/forecast datasets in a relatively well posed experimental framework, with careful analysis.

I see I am an additional reviewer, and that the previous reviewers have provided significant input - with the authors' replies indicating further improvement to the manuscript.

I find the paper, the associated hypothesis being tested and the experimental framework and analysis all sound. I think this work is important and timely, and recommend publication.

I only have a couple minor concerns.

I resonated with one of the reviewer comments regarding how much of this has previously been noted under different names, and the present just iterating on it with the nomenclature of ARs rather than somewhat generic fronts/cyclones. However, I am only so versed in the previous literature, so I hope the author's have treated previous literature appropriately. I do think there is value in casting this in terms of ARs given the significant interest in this area of study.

The authors make mention of how these results indicate some level of subseasonal predictability of ARs in the abstract but this is never really supported, investigated or analyzed. I think this

inference should be removed and a future study should perform a study focused on the predictability considerations, and how this actually manifests in terms of producing better forecasts at 2-4 week / 2 month time scales.

Page 2 line 48 - "Though forecasts of overall ... have b been improved in current weather forecasts and climate models..." how is this supported by references?

Page 2 line 52 - The reference about a community effort ... for AR science seems a bit overdone, the ARTMIP goals are limited to a subdomain of AR science - e.g. impacts of detection algorithms on various science questions, thus I think it is overstated. For a broad reference here and at the beginning, the authors might consider referencing the new Ralph et al. Springer book on ARs "Atmospheric Rivers", as well as the AR Recon program led out of Scripps that really looks at this problem of forecasting ARs as they make landfall on the west coast.

Reply to Referee #1

We would like to thank the referee again for the invaluable comments. We have carefully considered each of your comments (listed as bold and Italic below) and revised the manuscript accordingly. Below is a point-to-point reply to your comments:

The authors have satisfied my review points and I can recommend publication. However, I encourage them to also consider the following two additional comments before publication.

1. Response to request for PRISM data inclusion, Lines 122-123 original manuscript: Thank you for this explanation. Something similar should be noted in the paper itself because I think this will be a common questions amongst most readers.

A note has been added in the manuscript to explain why PRISM analysis is not included in the manuscript. See **Line 206-210**, it reads: “However, a similar analysis applied to the PRISM did not yield a statistically significant precipitation response. This may not be surprising because 1) PRISM only covers a limited area that is south of the most significant precipitation response to mesoscale SSTs and 2) the data records are relatively short and sample size is relatively small.”

2. Additional experiments as a response to Reviewer 2: I find the difference between the eddy-only and front-only experiments compelling and very intriguing! This is indeed a valuable addition to this work and I strongly recommend that it is a) included in the extended figures, in particular figure R7 and b) added as a subsection in the Methods section as supplemental material focused on isolating fronts vs eddies. I look forward to analysis on why the front-only experiment lead to a decrease in IVT and precipitation!

Following your advice, a subsection titled “Relative Importance of Ocean Eddies vs. SST Fronts in Driving the AR Response” is added in Methods. See **Line 445-478** and Supplementary **Fig. 8**.

Reply to Referee #3

We would like to thank the referee for the invaluable comments. We have carefully considered each of your comments, listed as bold and Italic characters below, and revised the manuscript accordingly. Below is a point-to-point reply to your comments:

This study examines the role of mesoscale SST eddies in the North Pacific on AR frequency and intensity - including landfalling IVT and precipitation. It tests the hypothesis that these mesoscale SST features impart a non-linear impact on the transport of moisture out of the PBL, that leads to greater moisture export out of the boundary layer into the free atmosphere in the region of cyclone development leading to increased AR frequency and intensity. They test this hypotheses with a number of modeling/forecast datasets in a relatively well posed experimental framework, with careful analysis.

I see I am an additional reviewer, and that the previous reviewers have provided significant input - with the authors' replies indicating further improvement to the manuscript. I find the paper, the associated hypothesis being tested and the experimental framework and analysis all sound. I think this work is important and timely, and recommend publication. I only have a couple minor concerns.

1. I resonated with one of the reviewer comments regarding how much of this has previously been noted under different names, and the present just iterating on it with the nomenclature of ARs rather than somewhat generic fronts/cyclones. However, I am only so versed in the previous literature, so I hope the author's have treated previous literature appropriately. I do think there is value in casting this in terms of ARs given the significant interest in this area of study.

We agree with the reviewer and reviewer #2 that many previous studies have investigated: (1) the influence of mesoscale SSTs in frontal regions (e.g., Kuroshio and Gulf Stream Extension) on extratropical cyclones, atmospheric fronts or precipitation and (2) the relationship between the occurrence of extratropical cyclones and ARs. Our research is motivated by these previous studies. We believe we have tried our best to survey the relevant literature in the paper to the extent of the limitation imposed by the journal (see **Line 93-101**). Furthermore, one of the novel findings of this study is that the ARs response to mesoscale SSTs is not necessarily associated with changes in extratropical cyclones. We believe this finding offers a new perspective on the relationship between ARs and extratropical cyclones.

2. The authors make mention of how these results indicate some level of subseasonal predictability of ARs in the abstract but this is never really supported, investigated or analyzed. I think this inference should be removed and a future study should perform a study focused on the predictability considerations, and how this actually manifests in terms of producing better forecasts at 2-4 week / 2 month time scales.

We agree. The sentence “implying the potential influence of the dynamical processes on AR predictability at subseasonal-to-seasonal time scales” in the abstract is now removed (See **Line 47**). In the main text, we also modify the sentence “This indicates that the influence of mesoscale SST forcing can occur on weekly time scales, thereby potentially affecting the predictability of AR-related heavy precipitation events along the West Coast of North America.” to “This indicates that the influence of mesoscale SST forcing can occur on weekly time scales, highlighting the need for future investigations

on whether the mesoscale SSTs can potentially affect the predictability of AR-related heavy precipitation events along the West Coast of North America on S2S time scales.” (See **Line 237-240**).

3. Page 2 line 48 - "Though forecasts of overall ... have b been improved in current weather forecasts and climate models..." how is this supported by references?

Wick et al. (2013) shows that the increased model resolution in current weather forecast models reduces bias of forecast ARs, but does not produce any significant improvement in forecasting the timing and location of landfalling ARs , highlighting the difficulty of current models in simulating landfalling ARs (Nardi et al. 2018). Forecasting AR related precipitation is proven to be even more challenging (Ralph et al. 2010; Lavers et al. 2014). We agree with the reviewer that the statement of “forecasts of overall occurrence and intensity of ARs have been improved” is not quite accurate. The revised sentence now reads: “Although increasing model resolution in current weather forecast models leads to forecast bias reduction of overall AR occurrence and intensity, the timing and location of landfalling ARs, as well as their precipitation impact, are notoriously difficult to predict⁷⁻¹¹”. One of the references was also updated. See **Line 65-68**.

4. Page 2 line 52 - The reference about a community effort ... for AR science seems a bit overdone, the ARTMIP goals are limited to a subdomain of AR science - e.g. impacts of detection algorithms on various science questions, thus I think it is overstated. For a broad reference here and at the beginning, the authors might consider referencing the new Ralph et al. Springer book on ARs "Atmospheric Rivers", as well as the AR Recon program led out of Scripps that really looks at this problem of forecasting ARs as they make landfall on the west coast.

We agree that Ralph et al.'s new book is a better reference to cite, containing a comprehensive collection of current understanding and advancement of AR science, and the ARTMIP goals are limited to evaluating impacts of AR-detection algorithms. The sentence is rephrased with additional references added. It now reads: "There is a concerted, ongoing research effort to understand and quantify predictability and uncertainty in forecasting AR¹²⁻¹⁶, including a community-driven project dedicated to evaluate impacts of AR detection algorithms on various science questions¹⁷⁻¹⁸." See **Line 69**.